# Beyond RNA Structure Alone: Complex-Aware Fusion for Tertiary Structure-based RNA Design

## Abstract

Tertiary structure-based RNA design plays a crucial role in synthetic biology and therapeutics. While existing methods have explored structure-to-sequence mappings, they focus solely on RNA structures and overlook the role of complex-level information, which is crucial for effective RNA design. To address this limitation, we propose the **C**omplex-**A**ware tertiary structure-based **R**NA **D**esign model, **CARD**, that integrates complex-level information to enhance tertiary structure-based RNA sequence design. To be specific, our method incorporates protein features extracted by a protein language model (e.g., ESM-2), enabling the design model to generate more accurate and complex relevant sequences. Considering the biological complexity of protein-RNA interactions, we introduce a distance-aware filtering for local features from protein representation. Furthermore, we design a high-affinity design framework that combines our CARD with an affinity evaluation model. In this framework, candidate RNA sequences are generated and rigorously screened based on affinity and structural alignment to produce high-affinity RNA sequences. Extensive experiments demonstrate the effectiveness of our method with an improvement of **7.3%** compared with the base model without our complex-aware feature integration.

## 1 Introduction

RNA plays a pivotal role in biological research, serving as a key molecule in various cellular processes. Beyond its natural functions, RNA design based on structures, also known as the inverse RNA folding problem, has become increasingly important across diverse fields, including synthetic biology, therapeutics, and biotechnology. By leveraging the 3D structure, researchers can engineer RNA molecules to perform specialized and targeted functions, driving innovation in these areas.

RNA molecules design based on structure has been a long-standing focus in computational biology, driven by its applications in synthetic biology, therapeutics, and biotechnology. Early efforts rely on stochastic optimization and energy-based optimization (Lorenz et al., 2011; Zuker, 2003) or physically informed optimization (Yesselman & Das, 2015). With the advancement of deep learning, this area has seen a shift toward more sophisticated computational approaches that can model the complex relationship between RNA sequences and their structures. For instance, RDesign Tan et al. (2024) and RhoDesign Wong et al. (2024) utilize graph modeling to encode the 3D structures and decode the sequence using GNN and Transformer, respectively. RDesign Tan et al. (2024) further leverages contrastive learning to achieve data-efficient representation learning.

However, these methods focus on modeling RNA structures alone, overlooking the crucial interaction dynamics in RNA complexes, as shown in Fig. 1 (a). As pointed out by Zhao (2024), different from protein folding which roughly follows Anfinsen's rule, RNAs are highly flexible and rely on interactions with proteins, DNA, and other biomolecules to achieve folding. Furthermore, RNAs may adopt various conformations by interacting with distinct macromolecules at different stages of functioning. Given that RNA structures are heavily influenced by interactions within complexes, analyzing RNA structures alone is insufficient for inverse folding. A comprehensive approach must consider the complex-specific interactions that model RNA design and functionality.

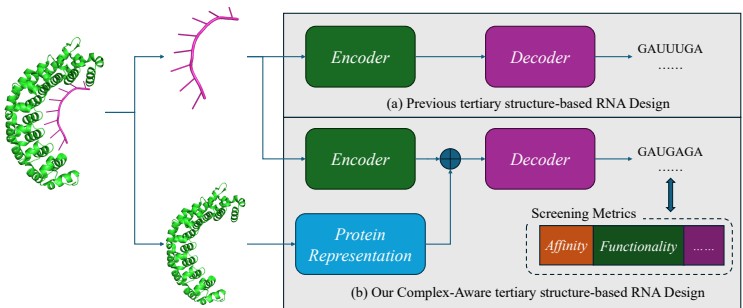

Figure 1: (a) Previous tertiary structure-based RNA design methods typically focus solely on RNA structural information. However, the critical interaction of RNA in complex has often been overlooked. (b) In this paper, we incorporate complex information into model to enhance its performance. Moreover, our model enables more flexible designs tailored to downstream requirements, such as focusing on functionality or affinity, thereby facilitating more targeted RNA design.

To address the above challenges, in this paper, we propose the **C**omplex-**A**ware tertiary structure-based **R**NA **D**esign model, **CARD**, that integrates RNA structural information and protein-RNA context to generate RNA sequences with complex context. In our CARD, apart from RNA tertiary structural representation, we utilize a pre-trained protein language model (PLM) to effectively represent the protein features, and fuse with the RNA representation to capture the complex interactions between RNA and proteins, as shown in Fig. 1 (b). Specifically, we adopt a Complex-Aware Transformer (CAFormer) to integrate RNA structural features and protein representations, enabling the generation of RNA sequences that adhere to key structural and complex context. Considering the biological reality that an RNA may interact with a protein through multiple binding sites and a single RNA can interact with different proteins, we introduce a distance-based stratification filtering to select the local and global representation for integration rather than using the representation of the whole protein sequence. This filtering can make the model focus more on the interaction regions and discard the effect of irrelevant regions. By combining the design model with a learning based affinity prediction model, our framework can screen the generated RNA sequences through an iterative fashion. Candidate sequences are first filtered based on affinity and then validated for structural compatibility using folding models, ensuring alignment with key metrics. By iteratively filtering, our framework can ensure the designed sequences have both high affinity and structural compatibility, offering a computationally efficient alternative to traditional experimental methods like SELEX.

In summary, our contributions are three-fold: (1) We propose the Complex-Aware tertiary structure-based RNA design framework, **CARD**, that integrates RNA tertiary structures and protein-binding contexts, enabling a complex-aware RNA design. (2) For complex information fusion, we introduce the Complex-Aware Transformer (CAFormer) for complex information integration and a distance-aware filtering for the local features of protein representation, enabling the model to capture the protein-RNA complex context and focus on interaction regions within complexes. (3) Extensive experiments on PRI30K and PRA201 (Han et al., 2024) datasets demonstrate the effectiveness of our framework. Additionally, the case study on high-affinity RNA design showcases the practical capacity and adaptability of our approach.

## 2 RELATED WORK

**Structure-to-Sequence RNA Design.** RNA design aims to generate sequences that fold into predefined structures. Early methods focused on secondary structure optimization, using thermodynamic parameters and energy minimization. Tools like RNAfold (Lorenz et al., 2011) and Mfold (Zuker, 2003) predict RNA secondary structures based on the minimum free energy principle. As understanding of RNA structure has advanced, focus has shifted to complex tertiary structure-based design due to RNA's high conformational flexibility, which challenges traditional thermodynamic methods (Ken et al., 2023). Recent deep learning approaches for RNA tertiary structure design include

gRNAde (Joshi et al., 2024), which utilizes geometric deep learning and graph neural networks to generate RNA sequences; RiboDiffusion (Huang et al., 2024), a diffusion model for inverse folding leveraging RNA backbone structures; RDesign (Tan et al., 2024), which employs a data-efficient learning framework with contrastive learning for tertiary structures; and Rhodesign (Wong et al., 2024), focusing on RNA aptamer design by guiding sequence generation through structural predictions. However, these methods often neglect the role of proteins in RNA design. Since RNA functionality depends not only on its structure but also on its interactions with proteins, we propose an RNA design approach that integrates protein information using protein language models.

**Protein-RNA Complex Analysis.** Protein-RNA interactions are crucial for gene regulation, RNA splicing, and stability. Various experimental techniques have been developed to characterize these interactions. Systematic Evolution of Ligands by EXponential Enrichment (SELEX) (Tuerk & Gold, 1990) is an in vitro method that selects high-affinity RNA aptamers for specific proteins. While useful for identifying RNA recognition motifs, it does not capture endogenous interactions in vivo. To address this, Crosslinking and Immunoprecipitation (CLIP) (Ule et al., 2005) employs UV crosslinking to stabilize RNA-protein complexes, followed by immunoprecipitation and sequencing. Variants such as HITS-CLIP (Licatalosi et al., 2008) and iCLIP (Konig et al., 2011) enhance resolution, while eCLIP (Van Nostrand et al., 2016) improves reproducibility and efficiency. RNA Immunoprecipitation Sequencing (RIP-seq) (Keene et al., 2006) provides a crosslink-free alternative, capturing RNA-protein interactions under physiological conditions, though with lower resolution. RNA Bind-N-Seq (RBNS) (Lambert et al., 2015) characterizes RNA-binding protein sequence preferences in vitro, revealing RNA recognition motifs. However, these methods cannot be applied to proteins where experimental data do not exist. This limitation highlights the need for computational approaches capable of generalizing across different protein-binding RNA.

**Affinity Evaluation.** The binding affinity between RNA and proteins is essential for validating RNA functionality. Current sequence-based prediction methods include PNAB (Yang & Deng, 2019), which manually extracts biochemical features and employs machine learning techniques like Support Vector Regression and Random Forests. DeePNAP (Pandey et al., 2024) utilizes convolutional neural networks to extract one-dimensional sequence features. Advanced approaches such as Pred-PRBA (Deng et al., 2019), PRdeltaGPred (Hong et al., 2023), and PRA-Pred (Harini et al., 2024) incorporate protein structural features—including secondary structures and binding interface characteristics like binding sites and hydrogen bonds—to achieve higher prediction accuracy with limited datasets. However, the scarcity of RNA-protein complex structures presents significant challenges. To address this, CoPRA (Han et al., 2024) consolidates a protein-RNA binding affinity dataset with about 300 complexes, PRA310, from multiple sources and introduces a multi-modal model that integrates RNA and protein large language models with comprehensive structural information.

## 3 METHOD

The overall pipeline of our framework is shown in Fig. 2. Our complex-aware RNA design model, CARD, leverages the pre-trained protein language model to extract the representation of protein in the complex, and leverages CAFormer to integrate the complex information into RNA representation for more effective inverse folding. Accomplished with a machine learning based affinity prediction model and structural prediction tools, a pure computational RNA sequence screening framework can be obtained for both task-specific requirements and structural compatibility based on protein-RNA complexes. This framework provides a scalable and efficient alternative to experimental techniques like SELEX, enabling diverse applications such as therapeutic RNA design and functional genomics.

### 3.1 RNA AND COMPLEX REPRESENTATION ENCODING

Given the protein-RNA complex, we first encode the RNA tertiary structures and protein representation, respectively. For the RNA structural representation, we utilize an architecture similar to RhoDesign (Wong et al., 2024) which uses Geometric Vector Perceptron (GVP) (Jing et al., 2020) to encode the tertiary structures. To be specific, the GVP processes the RNA backbone coordinates (i.e., C4', C1', N1) into vector and scalar features, integrating directional vectors and dihedral angles to capture structural information. This resulting representation, $h_{RNA} \in \mathbb{R}^{L_r \times D}$, is utilized for complex-aware feature fusion with the protein features. While for protein representation, protein sequences and structures are represented using pre-trained protein language models (PLMs), such as

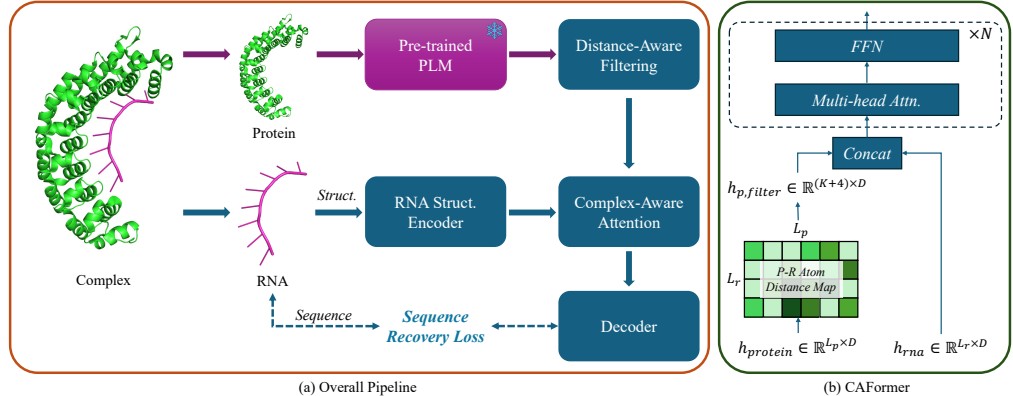

Figure 2: (a) The overall pipeline of our complex-aware tertiary structure-based RNA design model, CARD. We extract protein features by pre-trained PLM and integrate with RNA representation through CAFormer. (b) The detailed architecture of the CAFormer. The protein representation is first filtered via a distance-aware filtering to focus more on the protein-RNA interaction regions and then fused with the RNA feature through attention blocks.

ESM-2 (Lin et al., 2022) or ESM-3 (Hayes et al., 2025). These PLMs can extract high-dimensional embeddings, $h_{pro} \in \mathbb{R}^{L_p \times D}$, that capture sequence-level features, including secondary structure, functional motifs, and evolutionary relationships, providing a robust representation of the protein's sequence and structural information.

## 3.2 COMPLEX-AWARE FEATURE INTEGRATION

After obtaining RNA structural representation, $h_{RNA}$, and the representation of protein in complex, $h_{pro}$, we incorporate Complex-Aware Transformer (CAFormer) blocks to integrate complex-specific information into the RNA representation, enabling more effective design. The CAFormer architecture consists of a distance-filtering module, followed by $N$ Complex-Aware Attention blocks.

**Distance-aware Filtering.** To make the feature fusion pay more attention to the information within binding regions of the protein-RNA complex, we introduce a distance-filtering block. This block screens the protein representation, $h_{pro}$, based on the distances between protein amino acids and RNA nucleotides in the complex structure according to the biophysical principles. Specifically, we first select $K$ tokens corresponding to the closest amino acid-nucleotide pairs based on spatial proximity, prioritizing those with the shortest interatomic distances at the protein–RNA interface, as illustrated in Fig. 3a.

To further characterize the spatial context of the protein-RNA complex, we adopt a distance-based stratification model informed by structural statistics and biophysical considerations. Three concentric interaction shells are defined to reflect the progressive attenuation of molecular interactions and structural influence. The first shell (0–8 Å) captures direct atomic contacts (Hu et al., 2018; Krüger et al., 2018; Onofrio et al., 2014; Ferruz et al., 2021), including hydrogen bonds, hydrophobic interactions, and so on, which are critical for binding specificity and energetic stability. The second shell (8–16 Å) encompasses residues (Etheve et al., 2016; Salomon-Ferrer et al., 2013) that, while not in direct contact, may exert local influence through water-mediated bridges, electrostatic perturbations, or limited conformational coupling. The third shell (16–30 Å) accounts for distal residues (Said et al., 2021; Aguion et al., 2022) whose effects are mediated through global conformational rearrangements, long-range dipolar interactions, or allosteric regulation. We then partition the protein tokens according to their corresponding shell boundaries and apply average pooling within each shell to obtain three shell-level representations. Finally, to preserve the global information of the protein representation, we concatenate a global protein representation token by global average pooling (GAPool) with the locally selected tokens, resulting in the filtered protein representation, $h_{pro,filtered} \in \mathbb{R}^{(K+4) \times D}$.

$$D_{ij} = ||C_{p_i} - C_{r_j}||_2,$$

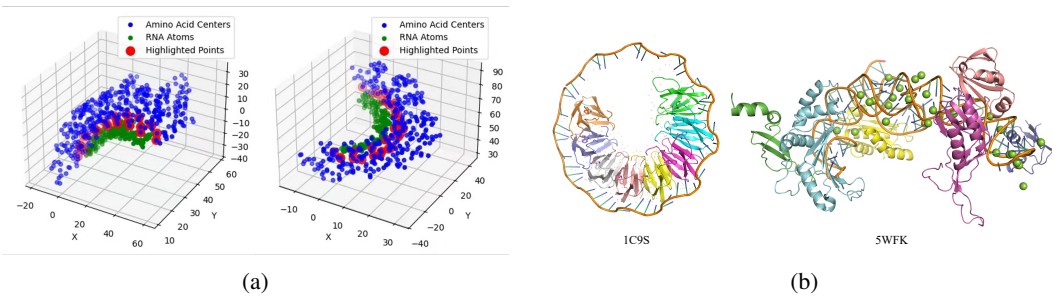

Figure 3: (a) An example of distance-aware filtering. Based on the distance between amino acids and nucleotides, we select the nearest $K$ amino acids to construct the protein representation, the selected indices are highlighted in red in the figure. (b) An example of protein-RNA multi interaction from 1C9S and 5WFK. Better viewing with color mode.

$$S_{local} = \text{argsort}(\min(D_{i,:}))[:K],$$
$$h_{pro,local} = \{h_{pro}^i | i \in S_{local}\},$$
$$h_{pro,shell} = \{\text{Pool}(h_{pro}^j, j \in \text{shell}_i), i = 1, 2, 3\},$$
$$h_{pro,global} = \text{GAPool}(h_{pro}),$$
$$h_{pro,filtered} = \text{concat}(h_{pro,local}, h_{pro,shell}, h_{pro,global}),$$

where $C_{p_i}$ and $C_{r_j}$ are the coordinates of the center of the $i$-th amino acid and $j$-th RNA nucleotide.

Notably, considering the biological complexity of protein-RNA interactions, where a protein may bind to an RNA through multiple binding sites (different poses), and also a single RNA can interact with different proteins, this filtering not only captures the global information of the protein within the complex but also enables the model to focus more on the features of the interaction regions. Fig. 3b gives two examples of protein-RNA multi interaction of a single RNA with the same protein and different proteins. We can integrate complex features more efficiently while mitigating the influence of non-interaction regions, thereby obtaining a more refined RNA representation.

**Complex-Aware Attention Block.** Given the RNA structural representation, $h_{RNA}$ and the filtered protein representation, $h_{pro,filtered}$, we first concatenate these two representations to form a unified complex representation, $h_{com} \in \mathbb{R}^{(L_r+K+4) \times D}$. We leverage $N$ multi-head self-attentions with $h_{com}$ as key and value, enabling the RNA representation to capture the relevant features from the protein context. This process enhances the RNA representation with complex-aware information, facilitating more effective RNA design.

$$h_{com} = \text{concat}(h_{RNA}, h_{pro,filtered}),$$
$$y = \text{Attention}(\text{Proj}_{q,k,v}(h_{com})) + h_{com},$$
$$\hat{h}_{com} = \text{FeedForward}(y) + y,$$

where $\text{Proj}_*$ represents the linear projection. For simplicity, here we omit the LayerNorm operations and output projection in the formulation. After applying the complex-aware attention, we extract the indices corresponding to the RNA representation, which serve as the final RNA representation for decoding operations. During training, the model is trained with teacher forcing using CE loss.

### 3.3 HIGH-AFFINITY RNA DESIGN FRAMEWORK

Building upon the inverse folding model detailed above, in this section, we present the high-affinity RNA Design framework, a comprehensive framework that integrates protein-RNA complex information and evaluation mechanisms. This framework is to design RNA sequences optimized under the constraint of high binding affinity and structural compatibility, ensuring suitability for diverse downstream applications. The design framework consists of two main components: the structure-to-sequence design model and evaluation tools.

**Design Phase.** In this stage, our complex-aware design model synthesizes candidate RNA sequences by integrating RNA tertiary structures with protein-RNA complex information. Utilizing pre-trained

protein language models (PLMs) and our CARD, the design model can effectively capture the intricate relationships between RNA and binding proteins, enabling the generation of RNA sequences tailored for complex constraints, such as binding sites and interactions.

**Constraint Evaluation.** Following the sequence generation, the constraint evaluation is utilized to filter the RNA sequences to meet the task-specific requirements while maintaining structural compatibility within the protein-RNA complex. This evaluation combines metrics for both affinity evaluation and structural similarity. For the affinity evaluation, we train an ensemble regression model on PRA201 (Han et al., 2024) to predict the affinity score based on protein and RNA sequences. While for structural evaluation, we calculate RMSD score by aligning the original structure with the predicted structure by prediction models, including AlphaFold3 (Abramson et al., 2024), Rho-Fold (Shen et al., 2024), and RoseTTAFold2NA (Baek et al., 2024), etc. It is worth noting that the affinity evaluation can be replaced with other functional metrics to adapt to different task-specific requirements, providing flexibility for various applications.

**Iterative Screening.** To achieve high-throughput screening while balancing computational efficiency, the searching process adopts a two-step evaluation. In each iteration, the pipeline first evaluates the binding affinity of the RNA candidates, selecting the top $10\% \sim 20\%$ of sequences based on their affinity scores. These selected candidates are then sent to folding models for structural validation to obtain the most optimal candidates. The optimal candidates obtained will serve as templates for the following iterations, where new RNA sequences are generated and re-evaluated through the same process. By iteratively updating the candidate pool, the framework ensures that the generated RNA candidates not only exhibit high affinity but also maintain structural compatibility by pure computational tools, which can offer the potential to assist or even replace time- and labor-intensive experimental techniques like SELEX in the field of RNA design.

## 4 EXPERIMENT

### 4.1 DATASET AND IMPLEMENTATION DETAILS

**Dataset.** To evaluate the effectiveness of our CARD, we curate our dataset from the PRI30K and PRA201 datasets proposed in Han et al. (2024). PRI30K is a non-redundant collection of protein-RNA interaction pairs, encompassing proteins with a maximum residue length of 750 and RNA sequences ranging from 5 to 500 bases ($5 \leq L_r \leq 500$). While PRA201 is compiled from PDB-bind (Wang et al., 2004), PRBABv2 (Hong et al., 2023), and ProNAB (Harini et al., 2022), encompassing 201 unique protein-RNA complexes. Each sample comprises a single protein chain and a single RNA chain. Here we directly utilize PRA201 as the blind test set for evaluation and downstream high-affinity RNA design studies. While for PRI30K, we first exclude the RNA chains having high similarity with those in PRA201 and over-length RNA chains for computational efficiency ($L_r > 128$), resulting in a dataset comprising 21,050 protein-RNA pairs and 2,309 unique RNA sequences. For unbiased model evaluation, we cluster the RNA chains based on different similarity metrics, then the dataset is randomly divided into training and test sets in an 80%-20% ratio based on the clustering.

**Training Details.** We train our model on two A100 GPUs for 200 epochs with a batch size of 48 (24 per GPU). In CAFormer, we set the number of attention blocks to $N = 6$ and the local filtered size of the protein representation to $K = 64$. The total training time is approximately 6 hours.

### 4.2 QUANTITATIVE COMPARISON OF COMPLEX-AWARE RNA DESIGN

**PRI30K Test Split.** Tab. 1 presents the comparison results on the split test set of PRI30K. Our method achieves consistent and significant improvements over all reproduced baselines. Our method achieves the highest recovery rate and macro F1 score among all length categories with an overall recovery rate of 61.28% and a macro F1 score of 0.6061. In particular, traditional sequence-based methods perform poorly, where SeqLSTM with a hidden size of 128 achieves a recovery rate of only 29.00%, 31.28%, and 30.94% on short, medium, and long sequence categories, which are slightly better than random guessing. While for StructGNN and GraphTrans which are designed for protein inverse folding, they achieve an overall recovery rate of about 37%. Compared to the second-best method RhoDesign, we provide an improvement of 2.34%, 8.06%, and 10.32% on short, medium, and long sequence categories, respectively. A similar trend also exhibits in Macro F1

Table 1: Comparison results on the test set of PRI30K. *: We reproduce all the other methods on our PRI30K training set except for RDesign, due to the unavailability of their training code or insufficient algorithmic details. Therefore, we directly utilized the public checkpoint.

| Method | Recovery Rate (%) | | | | Macro F1 | | | |
|---|---|---|---|---|---|---|---|---|
| | Short | Medium | Long | All | Short | Medium | Long | All |
| SeqLSTM(h=128) | 29.00% | 31.28% | 30.94% | 30.65% | 0.2814 | 0.3166 | 0.3012 | 0.2927 |
| SeqLSTM(h=256) | 29.02% | 31.04% | 29.26% | 30.16% | 0.2867 | 0.3117 | 0.2870 | 0.2935 |
| SeqRNN(h=128) | 28.00% | 29.14% | 30.29% | 29.11% | 0.2731 | 0.2967 | 0.2996 | 0.2819 |
| SeqRNN(h=256) | 29.22% | 31.45% | 31.26% | 30.86% | 0.2892 | 0.3208 | 0.3006 | 0.2988 |
| GraphTrans | 31.22% | 39.69% | 41.15% | 37.17% | 0.3136 | 0.3945 | 0.4101 | 0.3430 |
| StructGNN | 30.89% | 39.53% | 42.19% | 37.22% | 0.3061 | 0.3933 | 0.4206 | 0.3386 |
| RhoDesign | 56.18% | 54.92% | 49.07% | 54.00% | 0.5701 | 0.5505 | 0.4908 | 0.5581 |
| gRNAde | 34.70% | 36.20% | 35.15% | 35.48% | 0.3342 | 0.3642 | 0.3520 | 0.3568 |
| *RDesign** | 36.19% | 49.00% | 52.40% | 45.64% | 0.3574 | 0.4872 | 0.5224 | 0.4070 |
| CARD (Ours) | **58.52%** | **62.48%** | **63.09%** | **61.28%** | **0.5960** | **0.6239** | **0.6298** | **0.6061** |

Table 2: Comparison results on the PRA201.

| Method | Recovery Rate (%) | | | Macro F1 | | |
|---|---|---|---|---|---|---|
| | Short | Medium | All | Short | Medium | All |
| GraphTrans | 30.73% | 43.47% | 33.33% | 0.2940 | 0.4351 | 0.3020 |
| StructGNN | 30.94% | 41.16% | 33.33% | 0.3046 | 0.4131 | 0.3107 |
| RhoDesign | 55.95% | 57.54% | 56.39% | 0.5674 | 0.5722 | 0.5677 |
| gRNAde | 31.62% | 43.07% | 34.79% | 0.3039 | 0.4313 | 0.3133 |
| *RDesign** | 36.50% | 52.31% | 40.88% | 0.3537 | 0.5218 | 0.3658 |
| CARD (Ours) | **60.74%** | **70.44%** | **63.42%** | **0.5774** | **0.7005** | **0.5863** |

performance, our method achieves consistent improvements across all sequence length categories, with gains ranging from 0.03 to 0.14. Specifically, our model improves the overall macro F1 score by about 0.05. The overall results reinforce the effectiveness of our method of capturing complex-aware features into RNA design, leading to more reliable predictions.

**PRA201 Blind Test.** Tab. 2 presents a quantitative comparison of different methods on the PRA201. Our method consistently outperforms other methods on both recovery rate and macro F1 score. Compared to the second-best method, RhoDesign (Wong et al., 2024), we achieve an improvement of 4.79%, 12.90%, and 7.03%, respectively. Similarly, our method achieves the best overall macro F1 score of 0.5863, representing a 3.46% improvement over the second-best RhoDesign.

Fig. 4 gives a violin plot of the detailed distribution of recovery rates for different methods across two test sets. It can be observed that GraphTrans (Ingraham et al., 2019), StructGNN, and RDesign (Tan et al., 2024) exhibit wider and more dispersed distributions with a lower median recovery rate, indicating high variance and inconsistent performance across different sequence lengths. Overall, our method maintains a clear advantage over other methods. While other methods exhibit wider spread and lower overall recovery rates, our method produces a more compact and higher distribution shape, demonstrating better stability and reliability performance across different datasets. We also present more detailed results of our CARD under different similarity splits in the appendix.

## 4.3 Ablation Studies and Case Studies

**Impact of Block Numbers of CAFormer.** In this part, we conduct several experiments to figure out the impact of the number of complex-aware attention blocks as shown in the left part of Tab. 3. Without any complex-aware attention blocks, where the design model only consists of RNA structure encoder and decoder blocks, the overall macro F1 score is 0.5581. With two blocks of CAFormer, the score achieves an improvement of 0.0148, 0.0601, and 0.0366 in short, medium length categories, and the overall macro F1 score. This improvement demonstrates the effectiveness

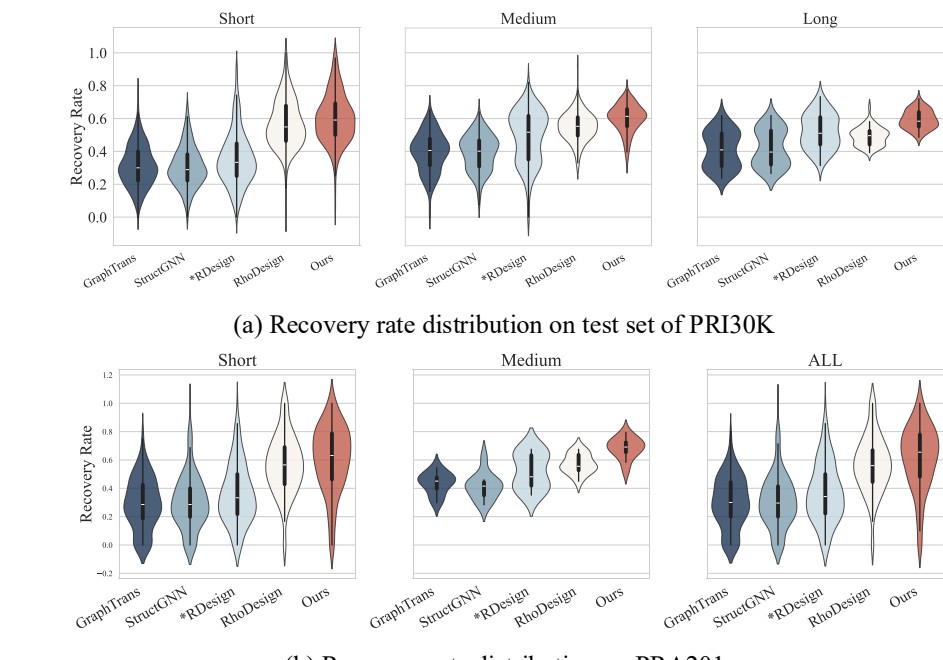

(a) Recovery rate distribution on test set of PRI30K

(b) Recovery rate distribution on PRA201

Figure 4: Violin plots for the recovery rate distribution of different methods across datasets and sequence categories.

Table 3: Ablation of number of blocks and the fashion of complex feature integration. Rand. and G.Dist. are random selection and greedy filtering, respectively. H.Dist. represents our stratification-based distance-aware filtering.

| Blocks | Macro F1 | | | | Global | Local | Recovery Rate | | |
|---|---|---|---|---|---|---|---|---|---|
| | Short | Medium | Long | All | | | Short | Medium | All |
| w/o Prot | 0.5701 | 0.5505 | 0.4908 | 0.5581 | ✓ | ✗ | 60.17% | 63.87% | 61.19% |
| $N = 2$ | 0.5849 | 0.6106 | 0.6218 | 0.5947 | ✓ | Rand. | 58.74% | 65.94% | 60.73% |
| $N = 4$ | 0.5845 | 0.6034 | 0.6016 | 0.5909 | ✓ | G.Dist. | **62.16%** | 68.73% | **63.98%** |
| $N = 6$ | **0.5960** | **0.6239** | **0.6298** | **0.6061** | ✓ | H.Dist. | 60.74% | **70.44%** | 63.42% |

of integrating complex information into RNA design. The performance consistently improves as the blocks increase. Our method achieves the highest macro F1 score with six blocks across different length categories.

**Impact of Protein Feature Filtering.** To figure out the best fashion of protein feature integration, we conduct several experiments as shown in the right part of Tab. 3. With the global representation by global average pooling of protein only, the model achieves an improvement of 4.8% on overall recovery rate. This result further reinforces the effectiveness of integrating protein context into the RNA design model. Applying random selection degrades the performance, with a drop of 1.43% on short sequences and 0.46% overall in recovery rate. Compared to greedy distance-based filtering, our stratification-based distance-aware approach further improves the recovery rate by 1.71% on medium, with minor drops on short and overall performance of -1.42% and -0.56%. We observe that our distance-aware filtering is more beneficial for longer RNA sequences, which we hypothesize is due to the fact that short sequences (<50nt) are predominantly determined by direct contact forces, whereas long-range dependencies play a more prominent role in shaping the structure of longer RNA.

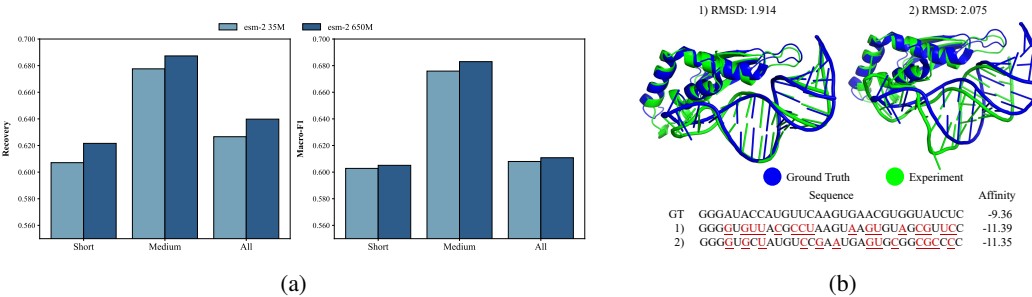

(a)                                                           (b)

Figure 5: (a) Performance of complex-aware attention integrating with representation produced by different ESM-2 models. (b) Structures and sequences of RNAs designed by CARD for 2LBS. The structures are predicted by AlphaFold3 and aligned with the native structure.

**Choices of Protein Representation.** In this part, we further conduct experiments using different sizes of ESM-2 to figure out the impact of protein representation models. The results are shown in Fig. 5a. The impact of the ESM-2 model size on performance is relatively minor. Using the 650M model for protein representation achieves an average improvement of only 1.25% in recovery rate and an average improvement of 0.0041 in macro F1 score compared to the 35M model.

**Case Study of RNA Design for 2LBS.** In this part, using 2LBS as an example, we demonstrate how our CARD can enhance the binding affinity of RNA to its target protein while maintaining the stability of the RNA structure using the high-affinity design framework described in Sec. 3.3. 2LBS is the solution structure of the double-stranded RNA binding domain of S. cerevisiae RNase III (Rnt1p) in complex with the AAGU tetraloop hairpin. We first input the native 2LBS complex into our CARD and perform RNA inverse folding to generate 1000 different RNA sequences. We then predict the binding affinities of these RNA sequences and select the top 20 sequences with the highest affinity. Next, we use AlphaFold3 (Abramson et al., 2024) to calculate the complex structures of these high-affinity RNAs with the target protein, and use these structures as inputs for the second round of design. For each complex, we generate 50 sequences, resulting in a total of 1000 sequences for the second round. This process is repeated to complete the third round of design. In Fig. 5b, we present two sequences from our design that exhibit higher binding affinity than the native RNA sequence, with structural differences from the native sequence being minimal. More details are provided in the appendix.

## 5   Conclusion and Limitation

In this paper, we propose a complex-aware tertiary structure-based RNA design framework, CARD, that integrates RNA structure representations and protein-binding contexts. Unlike existing approaches that focus solely on RNA structure, our framework explicitly incorporates protein-RNA interactions into the inverse folding process. To achieve this, we introduce the Complex-Aware Transformer (CAFormer) to fuse RNA and protein representations, ensuring RNA sequences adhere to both structural and complex constraints. Additionally, to make the model focus more on the interaction regions, we design a distance-aware filtering to select the protein representations for fusing rather than the whole protein representations. Our method further incorporates an iterative screening process that integrates an affinity predictor and structural validation, enabling efficient and scalable optimization of RNA sequences. Extensive experiments on the PRI30K and PRA201 datasets demonstrate the effectiveness of our approach. By integrating RNA tertiary structure modeling with protein-RNA interactions, our framework provides a robust and computationally efficient solution for RNA sequence design exploring the sequence space under diverse constraints.

While our framework demonstrates promising results, it still has two potential limitations. First, CARD relies solely on protein sequence features, overlooking explicit structural information that could provide richer geometric context for RNA design. Second, our CARD assumes a fixed complex structure and does not account for conformational flexibility or alternative binding states, which are prevalent in dynamic RNA–protein interactions. Future work may explore incorporating structural encoding and ensemble-based representations to address these limitations.

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

## ETHICS STATEMENT

This work develops a computational framework for complex based tertiary structure-based RNA design, especially for protein-RNA complexes, and is conducted entirely on publicly available datasets (PRI30K and PRA201 proposed in CoPRA), without involving human, clinical, or otherwise sensitive data. While the methods may contribute to advances in synthetic biology and therapeutics, any downstream applications must adhere to established biosafety standards and ethical guidelines. We are committed to transparency, fairness, and responsible stewardship in line with ICLR's Code of Ethics.

## REPRODUCIBILITY STATEMENT

Our experiments build on several open-source implementations, including CoPRA, RDesign, and RhoDesign, all of which are cited in the main paper. The training settings and hyperparameters are thoroughly described or analyzed in the experimental section to ensure reproducibility.

## USE OF LARGE LANGUAGE MODELS

In preparing this manuscript, large language models (LLMs) are used solely for polishing and refining the writing, and not for generating or proposing any methodological content.

# A  DATASET DETAILS

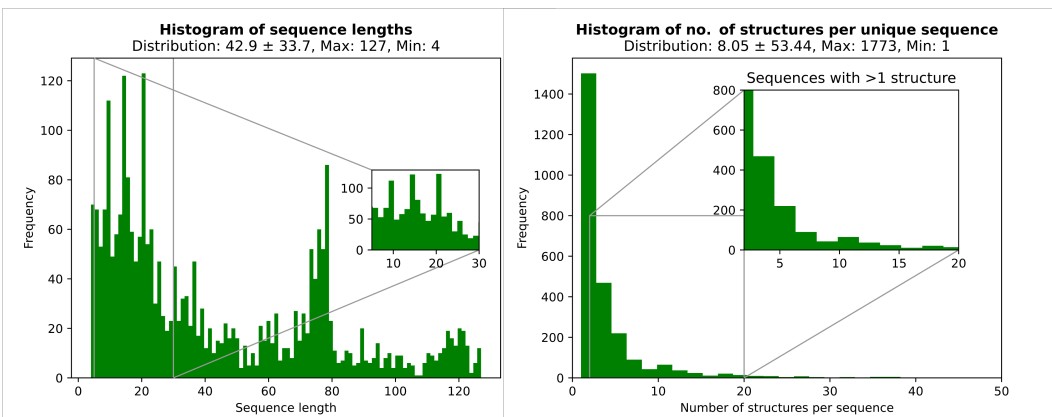

Figure 6: The detailed distribution over sequence length and protein-RNA pairs.

As mentioned in the main paper, we filter out the protein-RNA pairs from PRI30K to train our CARD. Fig. 6 shows the detailed distribution of our filtered sequences. The left part shows the frequency distribution of RNA sequence lengths in the dataset. The average sequence length is 42.9 with a STD of 33.7. Two peak regions can be observed in the length distribution, a primary peak in the $0 \sim 30$ nucleotide range and a secondary peak in the $60 \sim 80$ nucleotide range.

While the right part of the figure shows the histogram of protein-RNA pair counts per unique RNA sequence. This histogram exhibits a typical long-tail distribution, where the majority of RNA sequences are associated with only a small number of protein-RNA interaction structures, while a few sequences have an exceptionally large number of structures. The maximum number of structures per sequence is 1773. In Fig. 7, we also present a matched identity distribution of proteins based on sequence distribution in the filtered data. We further show the length distribution of PRA201 blind test set in Fig. 8. Most RNA sequences in the PRA201 blind test set fall into the short-length range. This dataset is mainly composed of short RNAs, with only a small portion of medium sequences.

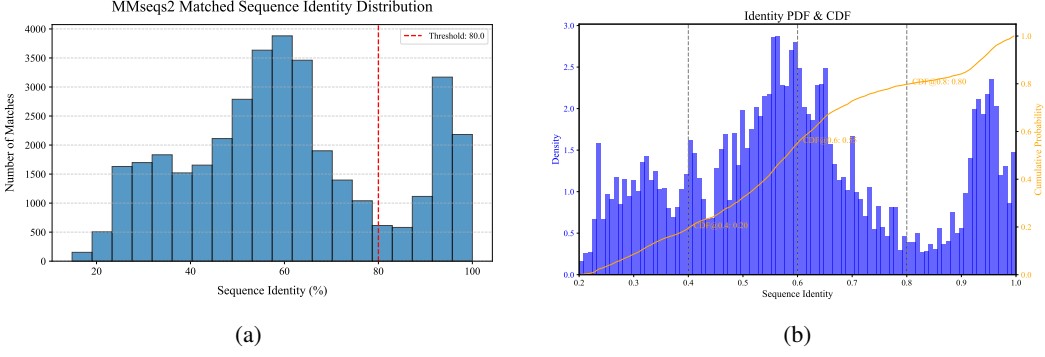

Figure 7: Protein sequence similarity distribution and CDF of filtered data.

# B  ADDITIONAL RESULTS UNDER DIFFERENT DATASET SPLIT

To evaluate the robustness of our model under different similarity conditions, we perform clustering based on both sequence and structural similarity. For RNA sequence similarity, we apply CD-HIT with identity thresholds of 30%, 50%, and 80%, where 80% is used as the default setting in the main experiments presented in the main text. For structural similarity, we use TM-score to cluster both RNA and protein structures, adopting a cutoff of 0.45 to distinguish structurally similar pairs, following the prior works. In addition, we also conduct an experiment based on protein structural

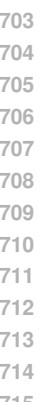

Figure 8: Length distribution of PRA201 blind test set.

Table 4: Performance under different similarity-based dataset splits. The results are evaluated on the PRA201 blind test set.

| Split | Recovery Rate | | | Macro F1 | | |
|---|---|---|---|---|---|---|
| | Short | Medium | All | Short | Medium | All |
| 30% RNA Seq. Sim. | 60.36% | 70.56% | 63.18% | 0.5740 | 0.7037 | 0.5834 |
| 50% RNA Seq. Sim. | 58.95% | 74.33% | 63.21% | 0.5661 | 0.7431 | 0.5789 |
| RNA TM-Score 0.45 | 59.83% | 68.98% | 62.17% | 0.5747 | 0.6859 | 0.5827 |
| 80% Prot Seq. Sim. | 61.60% | 73.60% | 64.92% | 0.6065 | 0.7331 | 0.6156 |

similarity. The results are presented in Tab. 4. Across all splits, our model maintains strong and consistent performance, with only minor fluctuations.

## C    AFFINITY PREDICTION MODEL

Table 5: 5-fold regression performance of our affinity predictor on PRA201. MAE is Mean Absolute Error, RMSE represents Root Mean Squared Error, PCC denotes Pearson Correlation Coefficient, and SCC is Spearman Correlation Coefficient.

| Fold | MAE | RMSE | PCC | SCC |
|---|---|---|---|---|
| 0 | 1.24 | 1.61 | 0.60 | 0.59 |
| 1 | 1.30 | 1.66 | 0.55 | 0.50 |
| 2 | 1.02 | 1.23 | 0.51 | 0.53 |
| 3 | 1.54 | 1.84 | 0.29 | 0.39 |
| 4 | 1.18 | 1.47 | 0.60 | 0.63 |
| Average | 1.26 | 1.56 | 0.51 | 0.53 |

As mentioned in the main text, we leverage a learning based affinity prediction model for high-throughput screening. In this part, we present how we conduct our affinity prediction model. Although CoPRA (Han et al., 2024) introduces an affinity prediction model by combining protein and RNA language models, it relies heavily on structural information, which contradicts our goal of achieving high-throughput screening. Furthermore, the structure predicted by folding models is often inaccurate, which can further negatively impact affinity predictions that depend on structural features. Notably, our goal is not to achieve SOTA performance on PRA201 but rather to design a robust affinity prediction model for our high-throughput screening framework.

To this end, we design a concise pure sequence based affinity predictor. Similar to CoPRA, we first extract protein sequence representation and RNA sequence representation by ESM-2 and RiNaLMo, and obtain the global representation via global average pooling. We then concatenate them to form

the overall sequence feature $h \in \mathbb{R}^{2560}$. While for the regression model, we employ an ensemble approach that integrates three models: LightGBM, XGBoost Random Forest (XGRF), and MLP. This ensemble leverages the strengths of each model to enhance predictive performance and robustness. The overall performance on PRA201 is shown in Tab. 5.

# D    CASE STUDIES FOR HIGH-AFFINITY RNA DESIGN

## D.1    MORE DETAILS FOR 2LBS

2LBS is the solution structure of the double-stranded RNA binding domain of S. cerevisiae RNase III (Rnt1p) in complex with the AAGU tetraloop hairpin. In this part, we show more details of the case study for 2LBS.

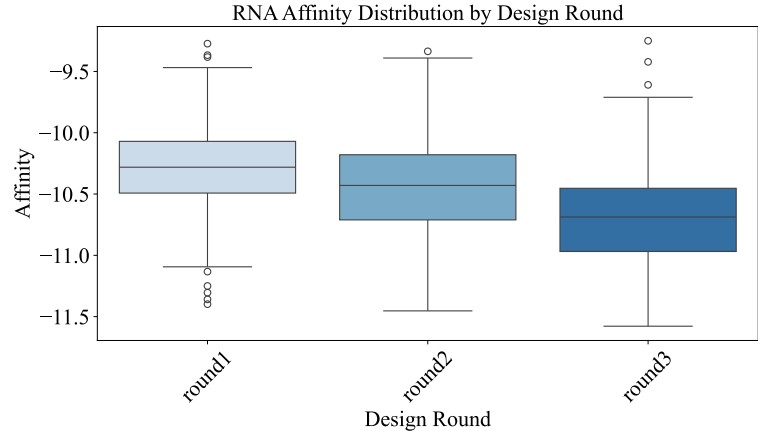

Figure 9: The distribution of predicted affinity across three design rounds.

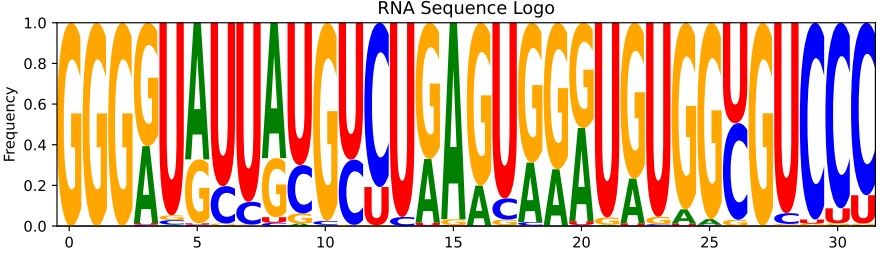

Figure 10: A sequence logo of generated RNA sequences.

We present the distribution of the binding affinity of designed RNA sequences in Fig. 9. This further validates the effectiveness of the CARD method in optimizing RNA binding affinity. Fig. 10 shows a sequence logo of an RNA sequence generated using Logomaker, which visually represents the composition and positional conservation in the RNA sequence. The vertical axis shows the frequency of $\{A, U, C, G\}$ at each position, while the height of each letter corresponds to its occurrence. It can be observed that certain positions are dominated by a relatively deterministic nucleotide, such as G at position $0 \sim 3$, GUCUGA at position $10 \sim 15$, and GUCCC at the end of the sequence. The conserved regions likely play essential roles in RNA stability or interactions, while the more variable positions suggest flexible or adaptive functional sites. The similar phenomenon is also reflected in Fig. 5b of the main text. The non-highlighted positions in the sequence and the interface regions in the folding visualization exhibit a relatively strong resemblance to the naive structure, indicating that these regions remain structurally stable.

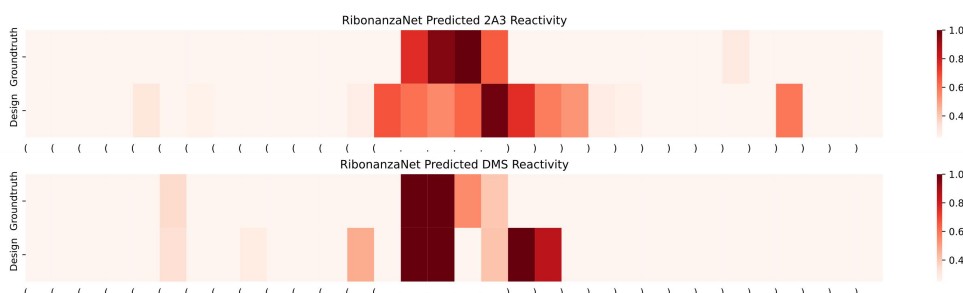

Figure 11: The detailed distribution over sequence length and protein-RNA pairs.

In Fig. 11, we further show the heatmaps of predicted chemical modification reactivity under 2A3 and DMS. Both 2A3 reactivity and DMS reactivity indicate that strong high-reactivity regions (deep red) are clustered in the central segment, particularly within the loop regions. The designed sequence and ground truth align well, showing that our CARD accurately captures chemical modification sensitivity.

## D.2 DESIGN FOR 2HGH

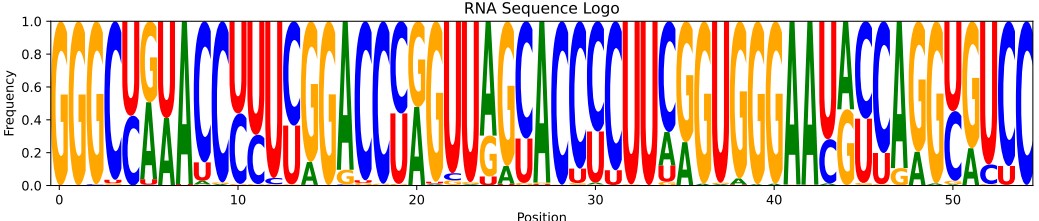

Figure 12: Sequence logo of generated RNA sequences for 2HGH.

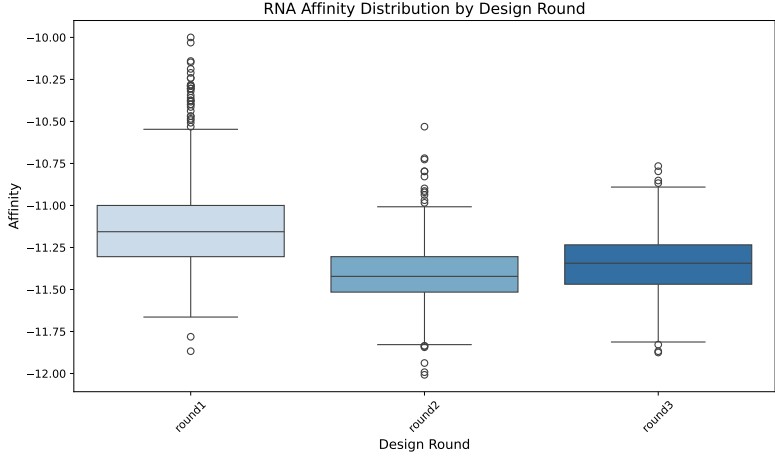

Figure 13: Affinity distribution of generated RNA sequences across rounds.

2HGH is the complex of transcription factor IIIA (TFIIIA) zinc fingers 4-6 bound to 5S rRNA 55mer, while TFIIIA is a Cys2His2 zinc finger protein that regulates the expression of the 5 S ribosomal RNA gene. Similar to the process of designing for 2LBS, we input the 2HGH complex into CARD to perform RNA inverse folding to generate 1000 RNA sequences, and then predict their binding

864
865
866
867
868
869
870
871
872
873
874

affinity. Next, we select the top 20 sequences with the highest predicted affinity and use AlphaFold3 to predict the structures, enabling a second-round design iteration. Through this iterative process, we generate 3 rounds of RNA sequences, progressively optimizing binding affinity for enhanced molecular interaction. We first show the sequence logo of generated sequences in Fig. 12. Although the RNA sequence length in 2HGH is longer compared to 2LBS, it retains more conserved regions. We conjecture that the structural constraints in 2HGH may lead to a higher degree of sequence preservation despite its increased length. The same phenomenon can also be shown in the affinity distribution of generated sequences across 3 rounds in Fig. 13. As observed in the figure, during the first two rounds, the high-affinity designs generated by CARD have already approached saturation. Consequently, in the third round, the newly generated sequences do not exhibit further improvements in affinity, and their distribution remains similar to that of the second round.

875
876
877
878
879
880
881
882
883
884
885
886
887
888
889
890
891

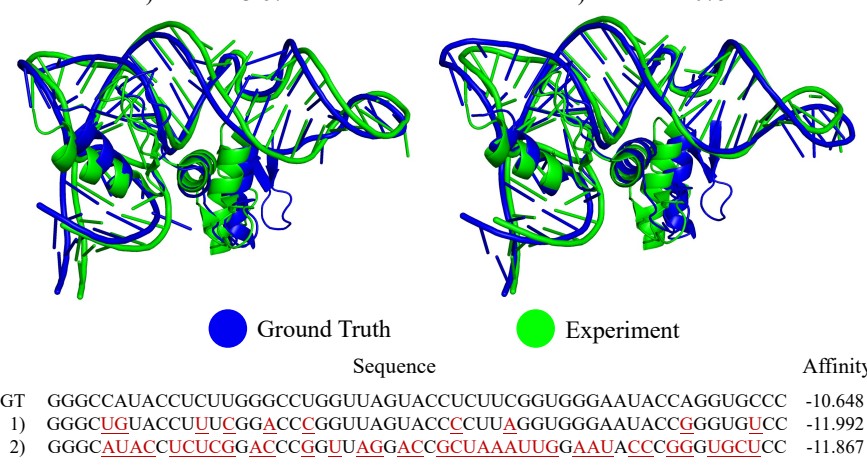

| | Sequence | Affinity |
|---|---|---|
| GT | GGGCCAUACCUCUUGGGCCUGGGUUAGUACCUCUUCGGUGGGAAUACCAGGUGCCC | -10.648 |
| 1) | GGGCUGUACCUUUUCGGACCCGGUUAGUACCCCUUAGGUGGGAAUACCGGGUGUCC | -11.992 |
| 2) | GGGCAUACCUCUCGGACCCGGUUAGGACCGCUAAAUUGGAAUACCCGGGUGCUCC | -11.867 |

892
893

Figure 14: Structures and sequences of RNAs designed for 2HGH.

894
895
896
897

Here, we select two candidates from round 2 and round 3 with the best trade-off between structural constraint and affinity improvement. The structure is predicted by AlphaFold3. The predicted structures and generated sequences are shown in Fig. 14.

898
899

# E EVALUATION OF STRUCTURAL CONSISTENCY

900
901
902
903
904
905
906
907
908
909
910
911
912
913
914

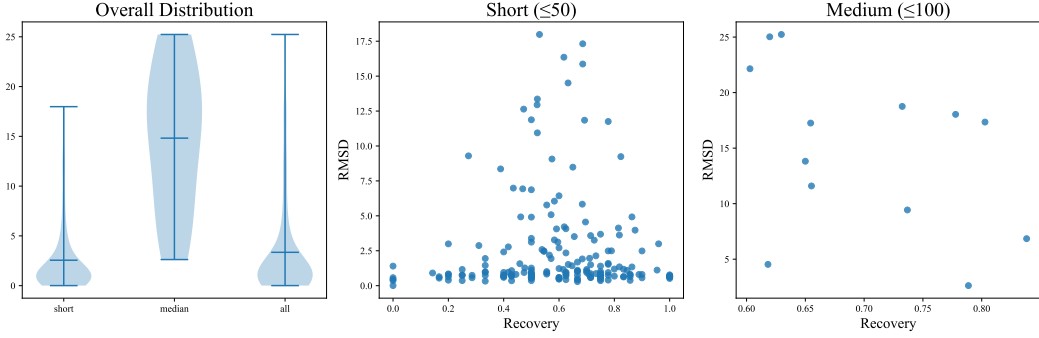

Figure 15: The distribution of RMSD of designed sequences.

915
916
917

For structural consistency evaluation, we fold all designed sequences in PRA201 blind test set using Protenix and compute backbone RMSD and TM-score against the native structures. Because this test set is dominated by short sequences, the overall RMSD remains low. On the full set, the RMSD ahieves a mean of 3.35 with a TM-score of 0.60. Short sequences show the most stable behavior,

with an RMSD of 2.56 (STD 3.55, N=188) and a TM-score of 0.62. For medium-length sequences, RMSD increases to 14.82 (STD 7.14, N=13). The overlla results of distribution boxplot and scatter plot of each length intervals are shown in Fig. 15.

# F    ADDITIONAL ABLATION RESULTS

Table 6: Additional ablation results with different settings. The results are evaluated on the PRI30K test split. *H* denotes stratification-based distance-aware filtering, *G* denotes greedy distance-based filtering, *D* denotes adding dihedral embeddings into PLM embeddings. *: During training with SaProt-650M, we observed that the public checkpoint on HuggingFace triggers the following warning, *Some weights of EsmForMaskedLM were not initialized from the model checkpoint at SaProt_650M_AF2 and are newly initialized: [..., 'esm.embeddings.position_embeddings.weight', ...].*

| Setting | Recovery Rate (%) | | | | Macro F1 | | | |
|---|---|---|---|---|---|---|---|---|
| | Short | Medium | Long | All | Short | Medium | Long | All |
| ESM 35M (H) | 57.66 | 60.26 | 61.75 | 59.72 | 0.5886 | 0.6025 | 0.6172 | 0.5947 |
| SaProt 35M (H) | 58.69 | 60.54 | 61.36 | 60.11 | 0.6008 | 0.6054 | 0.6135 | 0.6031 |
| ESM 650M (H) | 58.52 | 62.48 | 63.09 | 61.28 | 0.5960 | 0.6239 | 0.6298 | 0.6061 |
| *SaProt 650M** (H) | 58.41 | 60.14 | 59.11 | 59.32 | 0.5932 | 0.6012 | 0.5908 | 0.5950 |
| ESM 650M (G) | 58.96 | 60.12 | 59.39 | 59.56 | 0.6009 | 0.6007 | 0.5938 | 0.6002 |
| ESM 650M (H+D) | 59.60 | 60.41 | 60.10 | 60.06 | 0.6071 | 0.6043 | 0.6004 | 0.6058 |

**Comparison between ESM and SaProt.**    Tab. 6 reports the results of using ESM and SaProt as the protein representation backbone. Under the 35M setting, SaProt shows a small but consistent gain over ESM across all length ranges (e.g., 60.11% vs. 59.72% in overall recovery rate), suggesting that its structure-informed pretraining helps the model capture features relevant to protein–RNA binding. However, the 650M SaProt model performs noticeably worse than its ESM 650M counterpart. During training with SaProt 650M, the public checkpoint triggers a warning indicating that several weights, most importantly the `position_embeddings`, are missing and are newly initialized. Since position embeddings are essential for modeling order information, we conjecture this incomplete checkpoint is the main reason for the performance drop of SaProt 650M.

**Effect of distance-aware filtering schemes.**    Here we report the full comparison results between the stratification-based distance-aware filtering (*H*) and the greedy nearest-distance filtering (*G*) under the ESM 650M setting. The greedy strategy achieves slightly better accuracy on short RNAs (58.96% vs. 58.52%), which is reasonable because short RNA structures are dominated by local interactions where selecting the closest residues is sufficient. In contrast, the *H* method provides clear advantages on medium and long sequences (62.48% and 63.09% vs. 60.12% and 59.39%). Longer RNAs involve broader structural dependencies and benefit more from combining local and global shells, which *H* captures by design. This supports our motivation that stratified filtering better preserves complex-level context for larger structures.

**Adding structure modeling features.**    We further evaluate whether incorporating protein structural information helps the design model. For each protein backbone, we extract its dihedral angles, transform them into numerical features through sin/cos encoding, and map them to embeddings using a MLP. These embeddings are then added to the original sequence representation. As shown in Tab. 6, this naive integration does not improve performance (60.06% vs. 61.28% in overall recovery) and slightly reduces accuracy in medium and long ranges. This result suggests that simply injecting dihedral features into the token embeddings is not an effective way to leverage geometric signals, and more structured integration mechanisms may be needed.

Given the results from SaProt, we further analyze why directly injecting dihedral angle embeddings does not improve performance. SaProt 35M, which integrates structural knowledge at the pretraining stage, consistently outperforms ESM 35M, confirming that geometry is beneficial when incorporated into the representation space in a coherent manner. In contrast, post-hoc dihedral embedding

is applied after the protein tokens have already been filtered. This filtering breaks the geometric continuity that structural features naturally rely on, making it difficult for the added embeddings to be trained effectively. As a result, the naive addition of dihedral angle features not only fails to provide useful information but may also interfere with the pretrained sequence representation. These findings suggest that structural information needs to be introduced in a more unified way. A more suitable strategy is to employ pretrained models that jointly encode sequence and structure, such as SaProt, or leverage pretrained structural PLM, such as GearNet, to produce structure-aware tokens that can be fused with sequence tokens in a consistent geometric context. This approach preserves continuity in the structural space and is likely a more effective direction for incorporating geometric information into our framework.

