# OpenReview forum: "Beyond RNA Structure Alone: Complex-Aware Fusion for Tertiary Structure-based RNA Design"
_ICLR.cc/2026/Conference — Submitted to ICLR 2026_

### Official Review · Reviewer_tD58 · 2025-10-30

**Soundness:** 2
**Presentation:** 1
**Contribution:** 2
**Rating:** 2
**Confidence:** 4

**Summary:**

This paper introduces CARD (Complex-Aware tertiary structure-based RNA Design), a deep learning model for the RNA inverse folding problem. The authors rightly argue that existing methods, which focus solely on RNA structure, are insufficient because RNA folding and function are often dependent on interactions within a larger biomolecular complex (e.g., with proteins). The authors evaluate CARD on a filtered version of the PRI30K dataset and the PRA201 blind test set, showing good performance in sequence recovery rate compared to baselines like RhoDesign and RDesign.

**Strengths:**

1. Valid Problem Formulation: The paper's core premise is strong and addresses a well-known limitation in computational biology. Modeling RNA design in the context of its binding partners is a logical and necessary next step for the field, moving beyond the isolated "RNA-only" inverse folding problem.
2. Practical Framework: The proposed high-affinity design framework, which combines the generative model (CARD) with evaluative models (affinity predictor, structure folder), represents a practical, fully in silico workflow for optimizing functional RNA sequences.

**Weaknesses:**

1. Omission of Protein Structural Information: The model uses the 3D structure of the RNA (via GVP) and 3D distance information for its filtering mechanism, but it puzzlingly discards the 3D structure of the protein. Instead, it relies only on 1D sequence embeddings from a PLM (ESM-2). Given that protein-RNA interaction is fundamentally a 3D structural problem, this seems like a significant missed opportunity to provide the model with richer, more relevant features. The authors acknowledge this as a limitation but do not justify the initial design choice.
2. Focus on Sequence-Based Metrics: The evaluation relies heavily on sequence recovery rate and Macro F1-score. While the case studies show RMSD for specific examples, the paper would be more convincing if it included a comprehensive structural evaluation (e.g., average RMSD of folded designed sequences) across the entire test set. This would provide a clearer picture of whether the model's complex-aware approach also yields better structural preservation.
3. Purely In Silico Case Study: The high-affinity design case studies for 2LBS and 2HGH are entirely computational. The framework generates sequences with CARD and then "validates" them using AlphaFold3 for structure and a custom-trained ensemble model for affinity. This iterative loop optimizes for the predictors' biases, not for true in vitro binding affinity or structural stability. Without any experimental validation, this section only proves that the model can generate sequences that other computational models score highly, which is a form of circular logic.
4. Weak Justification for Core Novelty (Filtering): The main architectural novelty is the Distance-Aware Filtering module, specifically the "stratification-based distance-aware" approach ("H.Dist."). However, the ablation study in Table 3 does not provide a clear-cut case for its superiority.
  1. The simpler "G.Dist." (greedy distance filtering) achieves an overall recovery rate of 63.98%
  2. The proposed "H.Dist." achieves an overall recovery rate of 63.42%.
  3. The proposed method is outperformed by a simpler baseline in the main 'All' metric. The authors' explanation—that "H.Dist." is "more beneficial for longer RNA sequences"—feels like a post-hoc justification, especially when it underperforms on both "short" and "overall" recovery.
5. Poor Figure Quality: The figures throughout the paper (e.g., Figures 1, 2, 5, etc.) are simplistic and lack professional polish. They are not aesthetically pleasing. This detracts from the overall presentation quality of the manuscript and makes it harder to interpret the proposed architecture and results.

**Questions:**

1. Given that the simpler "G.Dist." filtering baseline outperforms your proposed "H.Dist." filtering in overall recovery rate (Table 3), can you provide a more robust justification for using the more complex, stratification-based method?
2. What was the rationale for discarding explicit protein 3D structural features and relying only on PLM sequence embeddings, especially when the RNA is represented structurally and your filtering mechanism is based on 3D distances?
3. In your high-affinity case study (Sec 3.3, Appendix D), you use an affinity predictor trained on the PRA201 dataset. Your case studies (2LBS, 2HGH) are also from PRA201. Was the affinity predictor trained on the entire PRA201 dataset, including 2LBS and 2HGH? If so, this seems to be a data leakage issue, where you are optimizing sequences for a predictor that has already been trained on the ground-truth affinity of the target complex.

---

> ### Author Response · Authors · 2025-11-24
>
> We thank the reviewer for the detailed feedback. Below we address each point.
>
> ## Omission of Protein Structural Information
>
> We appreciate the comment. In the revised version, we have added additional experiments exploring structure-aware tokens, including SaProt (structure-informed pretraining) and dihedral-angle embeddings (Appendix F). These results show:
> * Structure-aware pretraining (SaProt-35M) improves performance --> confirming that structural signals are useful.
> * Naively injecting structural features (e.g., dihedrals) slightly hurts performance --> because filtering breaks the geometric continuity (as we discussed in Appendix F).
> The reviewer suggests using a GVP-like encoder on the protein side. This is indeed a viable direction, but it requires joint structure-aware pretraining and retraining a full encoder, because only selected protein tokens enter the fusion module.
> Our dihedral ablation also shows that naive post-hoc injection is ineffective. Thus, adding a full GVP encoder without pretraining would be misaligned with the design.
> While integrating pretrained PLM with structural information, such as SaProt and GearNet, would provide a more suitable foundation for incorporating explicit geometry.
>
> ## Lack of structural evaluation
>
> Thank you for your advice. In Appendix E of revised version, we added a test-set structural evaluation on PRA201:
> * RMSD distribution
> * Scatter plots of recovery vs. RMSD
> * Violin plots
> These plots provide a comprehensive, dataset-level view of structural realizability.
>
> ## Weak justification for filtering module
>
> This misunderstanding arises from the highly skewed distribution of the PRA201 blind set:
> 188 out of 201 samples are short RNAs (<50nt).
> Thus, “All” is dominated by short RNAs, where greedy local selection naturally performs well.
>
> To address this concern, we added a complete evaluation on the PRI30K test split (Appendix F), which contains balanced sequence lengths. Results show:
> * G.Dist performs slightly better on short RNAs
> * H.Dist clearly outperforms on medium and long RNAs (where global context matters)
> This matches the expected biophysical rationale: local interactions dominate short RNAs; multi-shell contextual signals matter more for longer structures.
>
> We thank the reviewer for helping us clarify this point, which is now fully addressed.
>
> ## Poor figure quality
>
> We thank the reviewer for the comment. All figures will be refined to be aesthetically pleasing.
>
> ## Data leakage in affinity predictor training
>
> Our affinity predictor is used with 5-fold partitioning.
> For each test complex, we use the predictor from the fold where the complex is held out
> * e.g., if 2LBS belongs to Fold 0, we use the model trained on Folds 1/2/3/4
> * The predictor has never seen the complex or its label during training
> Thus, the affinity predictor strictly avoids leakage by design.

---

### Official Review · Reviewer_4Kce · 2025-10-30

**Soundness:** 3
**Presentation:** 2
**Contribution:** 2
**Rating:** 4
**Confidence:** 4

**Summary:**

I have reviewed this paper before. The paper tackles tertiary-structure-conditioned RNA inverse folding and argues that protein context matters. It encodes protein residues with a protein PLM (e.g., ESM-2), filters them by distance-aware selection, and (in the new version) aggregates them via concentric shells in addition to the usual top-K and a global token. These protein features are fused with RNA representations in a Complex-Aware Transformer (CAFormer). Beyond sequence recovery/F1 on PRI30K and PRA201, the paper adds an iterative high-affinity design loop using an affinity predictor and validates candidates by predicted complex structures(AF3 / RFNA) with RMSD checks. Overall, compared to the prior version, the method is more biologically motivated (multi-scale shells) and the narrative is cleaner. The results suggest consistent gains over structure-only baselines, and the design loop is closer to a practical pipeline.

**Strengths:**

1. The multi-scale, concentric-shell pooling (near/mid/far + global) rectifies the “narrow top-K only” view. It’s simple, implementable, and aligns with intuition about local vs. non-local protein influences.
2. Incorporating structure prediction (AF3 / RFNA) and an RMSD check into the screening loop moves beyond single-metric score chasing and acknowledges structural feasibility.
3. The paper presents a coherent CAFormer story and documents core training settings and data curation better than the earlier version.
4. The shells vs. alternatives and the small vs. large ESM-2 variants give readers knobs they can tune; even the observation that scale gains are modest is operationally useful.

**Weaknesses:**

- The paper demonstrates AF3/RFNA + RMSD usage in the loop and with case studies, but there is no large-scale, test-set-level structural analysis (e.g., RMSD/TM-score distribution across many examples, with mean/median/IQR and confidence intervals). This prevents a clean answer to: Does protein context improve the structural realizability of designed sequences at scale, not just in a handful of candidates?  Maybe, for a representative test split (or a sizable subset), report distributions of RMSD and/or TM-score for (i) RNA-only variant vs. (ii) protein-aware model, under the same AF3/RFNA inference settings. Include success-rate curves under varying RMSD/TM thresholds, and provide per-length (short/medium/long) breakdowns.


- The shells are distance-based pooling. There is still no explicit orientation / local frame/torsion angle encoding from the protein backbone, and no direct use of atomic coordinates beyond distances. If possible, consider adding a small ablation that augments the shell features with a lightweight geometric descriptor (e.g., local backbone frames, principal-axis orientation, dihedrals) at the selected residues. Show a couple of percent improvements or, if it doesn’t help, document the negative result transparently. I do think sole relying on PLMs is not a convincing way to support the authors' arguments.



- Minor:
The pipeline still depends on a learned affinity predictor trained on relatively small data. Maybe the author could elaborate more on that.

**Questions:**

Typos:
 sequences (¡50nt) , do you mean < ?

 Caption of Fig. 5,AlphaFold3 and aligned with the *naive* structure, maybe native is better?

---

> ### Author Response · Authors · 2025-11-24
>
> We sincerely thank the reviewer for the detailed and technically insightful feedback. Below we address each point.
>
> ## Need for structural analysis
>
> Thank you for raising this important point.
> In the revised version, we have added a structural evaluation in Appendix E. Specifically, we fold all designed sequences on the PRA201 blind test set using AlphaFold3 and report:
> * RMSD scatter plots vs. recovery
> * Violin plots of RMSD distributions
> These results provide a more complete results of structural evaluation, not only for selected case studies.
>
> ## Missing geometric descriptors beyond distances
>
> We appreciate this suggestion. In Appendix F (revised version), we expanded our analysis to include this exact concern. We evaluated two types of structure-informed signals:
> * SaProt embeddings (structure-aware during pretraining)
> * Dihedral-angle embeddings injected on top of ESM features
> As summarized, SaProt improves performance at the 35M scale, demonstrating that structure-aware pretraining is indeed helpful. However, directly injecting dihedral embeddings at the fusion stage did not improve performance, and in some cases degraded it. Our analysis indicates two main reasons:
> * Protein tokens are filtered before fusion, breaking geometric continuity, which makes post-hoc structural embeddings difficult to learn.
> * Structural features require a unified geometric context; naive addition after filtering is insufficient.
> These findings indicate: introducing geometry only works when it is integrated consistently at the pretraining stage, rather than appended after filtering. This motivates future extension toward models such as SaProt or GearNet that natively support structure-aware tokenization.
>
> ## Concerns about the learned affinity predictor
>
> We agree and provide additional clarification. PRA201(1to1 protein-RNA complex data) and PRA310(NtoN protein-RNA complex data)[1] are currently the largest available protein–RNA affinity datasets, but the sample size remains limited.
> We actively investigated alternative filtering strategies. Metrics such as iptm or plddt (from AF2/3) are often used for structural confidence estimation, but they require explicit folding of each protein–RNA complex. Folding hundreds or thousands of candidates per iteration is computationally prohibitive and fundamentally incompatible with high-throughput screening.
> Thus, a pure sequence-based affinity evaluator is required as the first-stage filter.
> Furthermore, Boltz2 has not been updated in Boltz codebase currently.
> We benchmarked our sequence-only ensemble predictor against the sequence-based components of reported results. Our model achieves competitive or better performance among sequence-only baselines:
>
> | Method | Struct | Seq | LM | PCC | SCC |
> | :-: | --- | --- | --| --- | --- |
> | LM+LR |X|Y|Y| 0.37 | 0.36 |
> | LM+RF |X|Y |Y|0.44 | 0.47 |
> | LM+MLP |X|Y |Y|0.40 | 0.41 |
> | LM+SVR |X|Y |Y|0.45 | 0.46 |
> | LM+Transformer |X|Y |Y|0.49 | 0.49 |
> | CoPRA |Y|Y|Y|0.57|0.59|
> |Ours|X|Y|Y|0.51|0.53|
>
> CoPRA does achieve higher performance, but only by using explicit structure of both protein and RNA. Integrating such structure-dependent predictors into our pipeline requires folding every generated protein–RNA complex, which contradicts our goal of high-throughput screening, and is further limited by the imperfect accuracy of RNA folding in AF3, as stated in its original report.
> Since no reliable sequence-only evaluator currently exists, we chose to build one from scratch, focusing on robustness and speed.
>
> [1] Han R, et al. Copra: Bridging cross-domain pretrained sequence models with complex structures for protein-rna binding affinity prediction [C] AAAI2025.
>
> ## Typos
>
> Thank you for pointing them out. These have been corrected in the revised version. One issue was due to math-symbol rendering, and another was a minor spelling error.

---

> > ### Comment · Reviewer_4Kce · 2025-11-27
> >
> > Thanks for the rebuttal; my concerns have been largely resolved, thus I have increased my score to 6.

---

### Official Review · Reviewer_3Qg2 · 2025-10-30

**Soundness:** 3
**Presentation:** 3
**Contribution:** 3
**Rating:** 4
**Confidence:** 3

**Summary:**

This paper introduces  a novel Complex-Aware RNA Design framework that addresses a critical limitation in existing methods by explicitly incorporating the protein-RNA interaction context into the inverse folding process. The core technical contributions include the Complex-Aware Transformer (CAFormer) and a biophysics-based distance-aware filtering mechanism. The paper further proposes a robust, scalable high-affinity design pipeline integrating sequence generation, affinity prediction, and structural validation, offering a computational alternative to resource-intensive experimental screening. Benchmarking against current baselines on challenging datasets (PRI30K, PRA201) demonstrates its superior performance and effectiveness.

**Strengths:**

1. The paper is the first to systematically and successfully use the complete "protein-RNA complex" as the core design context, rather than the isolated RNA structure. This shift aligns far better with biological reality.

2. The integrated high-affinity design framework seamlessly connects sequence generation, affinity evaluation, and structural validation. This comprehensive approach provides a powerful and computationally scalable solution for identifying functional sequences.

3. The method demonstrates performance improvements on both the standard benchmark (PRI30K) and, critically, on the more challenging blind test set (PRA201), proving the method's effectiveness.

**Weaknesses:**

1. The distance-aware filtering mechanism, while conceptually well-grounded in biophysics, empirically yields a slightly lower overall recovery rate than the simpler greedy distance selection (as shown in Table 3). The authors could address this trade-off by providing a deeper mechanistic explanation for this counter-intuitive result and clearly justifying the necessity of the complex biophysics-based approach when a simpler strategy performs better empirically.

2. The affinity predictor, a cornerstone of the high-affinity screening pipeline, is currently only validated through internal cross-validation. A comprehensive external benchmark against contemporary state-of-the-art affinity prediction models is crucial to objectively establish its competitiveness

3. The significant performance drop observed in Fold 3 compared to other cross-validation folds strongly suggests model sensitivity to data partitioning or distribution differences. This instability is concerning as it could introduce large, undesirable variance when the framework is applied in high-throughput, real-world screening contexts.

**Questions:**

1. Could the authors elaborate on how the ensemble scheme specifically leverages the "respective advantages" of the three lightweight models to ensure high screening accuracy? Furthermore, what is the rationale for not integrating a single, proven, high-accuracy predictor (such as Boltz2) directly into the pipeline, and have you benchmarked the resulting screening performance difference?

2. Given that interface geometric complementarity is crucial for protein-RNA recognition, the current use of ESM-2 only encodes protein sequence information. Have the authors considered integrating models (such as SAProt) that explicitly utilize both protein sequence and structural information to potentially enrich the binding mode representation and further enhance the design fidelity?

All other questions reiterate the items listed under Weaknesses. I am very willing to increase my overall score if the authors successfully address these concerns.

---

> ### Author Response · Authors · 2025-11-24
>
> We thank the reviewer for the detailed and constructive comments. Below we address each concern point-by-point.
>
> ##  Distance-aware filtering vs. greedy selection
>
> This difference arises from the composition of the PRA201 blind test set. As shown in Appendix Fig. 8 of revised version, 188 out of 201 samples fall in the short-length range. This distribution causes the evaluation of medium/long categories to be statistically insufficient, biasing the “overall” score toward the short-range regime.
> To clarify, we added the full comparison on the PRI30K test split in Appendix F, where all length ranges are well represented. The results show a consistent trend:
> * Greedy filtering performs slightly better on short RNAs, where interactions are dominated by local contacts.
> * Stratification-based filtering produces noticeably better results on medium and long RNAs, where global and multi-shell interactions matter more.
> This aligns with the expected behavior of both strategies:
> * Short RNAs rely primarily on nearest-neighbor interactions --> greedy works well.
> * Longer RNAs require multi-level context --> stratified shells become necessary.
>
> ## External benchmark for affinity predictor
>
> We appreciate this suggestion. The recent datasets PRA201 and PRA310[1] are indeed the largest currently available for protein–RNA affinity. We benchmarked our sequence-only ensemble predictor against the sequence-based components of reported results. Our model achieves competitive or better performance among sequence-only baselines:
>
> | Method | Struct | Seq | LM | PCC | SCC |
> | :-: | --- | --- | --| --- | --- |
> | LM+LR |X|Y|Y| 0.37 | 0.36 |
> | LM+RF |X|Y |Y|0.44 | 0.47 |
> | LM+MLP |X|Y |Y|0.40 | 0.41 |
> | LM+SVR |X|Y |Y|0.45 | 0.46 |
> | LM+Transformer |X|Y |Y|0.49 | 0.49 |
> | CoPRA |Y|Y|Y|0.57|0.59|
> |Ours|X|Y|Y|0.51|0.53|
>
> CoPRA does achieve higher performance, but only by using explicit structure of both protein and RNA. Integrating such structure-dependent predictors into our pipeline requires folding every generated protein–RNA complex, which contradicts our goal of high-throughput screening, and is further limited by the imperfect accuracy of RNA folding in AF3, as stated in its original report.
> Thus, for fast, scalable design, a robust sequence-only predictor is more appropriate.
>
> ## Performance drop in Fold 3
>
> This is a valid concern. The performance drop originates from label distribution imbalance in PRA201. The dataset was originally partitioned based on structural similarity, resulting in Fold 3 containing many complexes with extreme affinity values (e.g., > –6 or < –12), whereas most samples lie in the –8 to –11 range. This skew causes higher prediction difficulty.
> To avoid this bias, in our simulation studies we intentionally exclude Fold 3 from downstream validation, and instead use complexes from Fold 0 and Fold 4, where the affinity predictor for that fold is more stable (PCC/SCC >= 0.6).
>
> ## Integrating Boltz2/other tools
>
> We indeed explored the possibility of integrating a single high-performing model. However, several practical constraints make this infeasible in our high-throughput design setting.
> Although a technical report is published, the official codebase has not yet updated training and evaluation code for Boltz2. The current Boltz1 only supports protein–ligand affinity.
> We actively investigated alternative filtering strategies. Metrics such as iptm or plddt (from AlphaFold2/3) are often used for structural confidence estimation, but they require explicit folding of each protein–RNA complex. Folding hundreds or thousands of candidates per iteration is computationally prohibitive and fundamentally incompatible with high-throughput screening.
> Thus, a pure sequence-based affinity evaluator is required as the first-stage filter. However, as shown in Table above, existing sequence-only affinity predictors perform inconsistently and show limited correlation. Since no reliable sequence-only evaluator currently exists, we chose to build one from scratch, focusing on robustness and speed.
>
> ## Incorporating SaProt
>
> We agree, and we have added experiments in Appendix F to evaluate SaProt embeddings. SaProt 35M indeed outperforms its ESM counterpart, confirming that structure-aware pretraining is beneficial.
> However, we also found that post-hoc addition of structural signals, such as dihedral embeddings, does not help. As explained in the appendix, this is because:
> * Protein tokens are filtered before fusion, breaking geometric continuity
> * Naively adding structural embeddings after filtering prevents effective learning
> Therefore, the correct way to utilize structural signals is to rely on unified structure-aware pretraining, not post-hoc injection.
> Models such as SaProt or GearNet are promising next-step extensions. We highlight this direction in the revised discussion.
>
> [1] Han R, et al. Copra: Bridging cross-domain pretrained sequence models with complex structures for protein-rna binding affinity prediction [C] AAAI2025.

---

### Official Review · Reviewer_zkxc · 2025-10-31

**Soundness:** 4
**Presentation:** 4
**Contribution:** 4
**Rating:** 8
**Confidence:** 5

**Summary:**

This paper presents CARD (Complex-Aware tertiary structure-based RNA Design), a framework for RNA inverse folding that explicitly incorporates protein–RNA complex information rather than relying solely on RNA structures. The authors introduce a Complex-Aware Transformer (CAFormer) that fuses RNA structural features with protein representations from pretrained protein language models (e.g., ESM-2), enhanced by a distance-aware filtering mechanism to focus on interaction regions. Furthermore, they design a high-affinity RNA design framework combining CARD with an affinity evaluation model for iterative screening. Experiments on the PRI30K and PRA201 datasets demonstrate significant improvements over previous structure-only baselines (e.g., RhoDesign, RDesign), with clear gains in recovery rate and F1 score. Overall, the work contributes an effective and biologically meaningful approach to complex-aware RNA design and shows strong empirical results supported by detailed ablation and case studies.

**Strengths:**

The paper addresses an important and underexplored problem—RNA design within protein–RNA complexes—with a clearly novel formulation. The proposed CARD framework is original in integrating tertiary RNA structures with protein context via a Complex-Aware Transformer and distance-aware filtering, yielding clear methodological innovation. The work is technically sound, well-executed, and supported by strong empirical evidence across multiple datasets. The presentation is clear and well-organized, and the results demonstrate meaningful biological significance and consistent improvements over prior RNA design methods.

**Weaknesses:**

While the paper is well executed, a few areas could be strengthened. First, the method relies mainly on protein sequence embeddings without explicitly modeling protein 3D structures, which may limit its ability to fully capture geometric interaction patterns.

**Questions:**

None

---

> ### Author Response · Authors · 2025-11-24
>
> We sincerely thank the reviewer for the positive evaluation and for raising this insightful point. We fully agree that incorporating explicit protein 3D structural information is an important direction, and we have conducted additional experiments in the rebuttal revision to further examine its impact.
>
> In Appendix F, we evaluated two forms of structural modeling:
> * SaProt embeddings, which introduce structure-aware representations through pretraining; and
> * Dihedral-angle embeddings added on top of ESM features.
>
> The results show that SaProt-35M consistently outperforms ESM-35M, confirming that structure-aware pretraining is indeed beneficial for our task. In contrast, directly injecting dihedral features into ESM does not improve performance and in some cases degrades it. As discussed in the appendix, this is largely due to the fact that our protein token filtering is performed before fusion. This filtering breaks the geometric continuity that structural features depend on, making post-hoc structural embeddings difficult to optimize. These findings help clarify why a naive feature-level addition of 3D geometry is insufficient.
>
> Taken together, our results indicate that the key to effectively leveraging structural signals is to introduce them at the pretraining stage, where sequence and structure can be learned jointly and consistently. Approaches such as SaProt, GearNet, or other structure-informed encoders provide a more suitable foundation for incorporating explicit geometry, and exploring this direction is a natural extension of our framework.

---

### Author Response · Authors · 2025-11-24

We sincerely thank the reviewers for their time and thoughtful feedback. We carefully revised the manuscript to address the key concerns raised, and the major improvements are summarized below.

1. Structural Evaluation (RMSD/TM-score)

To address concerns regarding structural realizability, we added a test-set-level structural analysis in Appendix E:
Folded all designed sequences in the PRA201 blind set using Protenix.
Reported RMSD scatter plots, RMSD distribution (violin plots).

2. Additional Ablation Studies on Structural Features and Filtering

We expanded Appendix F with new ablations to clarify the role of structural signals and filtering mechanisms:
* SaProt vs. ESM: Demonstrated that structure-aware pretraining (SaProt-35M) improves performance, supporting the usefulness of protein structural information.
* Dihedral-angle embedding: Added a controlled ablation showing that naive post-hoc geometric feature injection does not help and can degrade performance, explaining the architectural constraints related to token filtering.
* H.Dist vs. G.Dist filtering: Provided a complete evaluation on the balanced PRI30K test split, confirming that stratified distance-aware filtering benefits medium/long RNAs, while greedy filtering is only advantageous in short-RNA–dominated settings.

---

### Meta-Review · Area_Chair_RiUK · 2026-01-06

**Summary:**

Majority of the reviewers are concerned about whether protein context improve the structural realizability of designed sequences at scale, rather than a few case studies. They are also concerned that the protein-RNA complex information is inadequately modeled since the protein encoding mainly comes from protein language model, which does not model the relative position between protein and RNAs.

**Reviewer Concerns:**

I don't think reviewer concerns are adequately addressed.

**Reviewer Scores:**

I think the reviewers will remain their scores even if they participated the discussion.

---

### Decision · Program_Chairs · 2026-01-26

Reject